

# How well do force fields capture the strength of salt bridges in proteins?

Mustapha Carab Ahmed[1], Elena Papaleo[1,2] and Kresten Lindorff-Larsen[1]

[1] Structural Biology and NMR Laboratory, Linderstrøm-Lang Centre for Protein Science, Department of Biology, University of Copenhagen, Copenhagen, Denmark
[2] Computational Biology Laboratory, Danish Cancer Society Research Center, Copenhagen, Denmark

## ABSTRACT

Salt bridges form between pairs of ionisable residues in close proximity and are important interactions in proteins. While salt bridges are known to be important both for protein stability, recognition and regulation, we still do not have fully accurate predictive models to assess the energetic contributions of salt bridges. Molecular dynamics simulation is one technique that may be used study the complex relationship between structure, solvation and energetics of salt bridges, but the accuracy of such simulations depends on the force field used. We have used NMR data on the B1 domain of protein G (GB1) to benchmark molecular dynamics simulations. Using enhanced sampling simulations, we calculated the free energy of forming a salt bridge for three possible lysine-carboxylate ionic interactions in GB1. The NMR experiments showed that these interactions are either not formed, or only very weakly formed, in solution. In contrast, we show that the stability of the salt bridges is overestimated, to different extents, in simulations of GB1 using seven out of eight commonly used combinations of fixed charge force fields and water models. We also find that the Amber ff15ipq force field gives rise to weaker salt bridges in good agreement with the NMR experiments. We conclude that many force fields appear to overstabilize these ionic interactions, and that further work may be needed to refine our ability to model quantitatively the stability of salt bridges through simulations. We also suggest that comparisons between NMR experiments and simulations will play a crucial role in furthering our understanding of this important interaction.

Corresponding author
Kresten Lindorff-Larsen,
lindorff@bio.ku.dk

## INTRODUCTION

Proteins are stabilized via the concerted action of numerous weak forces including those that arise from hydrogen bonds, the hydrophobic effect and salt bridges (*Dill, 1990*; *Zhou & Pang, 2017*). While we now know much about the relative contributions and physical origins of these effects, we still do not have quantitative models that allow us, for example, to predict accurately the overall stability of a protein given its three-dimensional structure. A quantitative understanding of protein stability would aid both to design proteins with improved stabilities (*Foit et al., 2009*), as well as understand how loss of protein stability may give rise to disease (*Casadio et al., 2011*; *Nielsen et al., 2017*).

We here focus our attention on salt bridges in proteins, where they occur between oppositely charged ionisable groups, most commonly between the negatively charged side chains of aspartate and glutamate or the carboxy-terminus, and positive charges in arginine, lysine or histidine side chains, or with the N-terminal ammonium group. A salt bridge is formed when the two residues involved are spatially close enough to form energetically favourable electrostatic interaction between the (partially) charged atoms (*Barlow & Thornton, 1983*). Although the first quantitative models of protein electrostatics are almost one hundred years old (*Linderstrøm-Lang, 1924*), our ability to predict electrostatic properties remain incomplete. Indeed, while ionic interactions were also early suggested to contribute substantially to the stability of certain proteins (*Speakman & Hirst, 1931*), their importance and energetic contribution was already controversial before they had been observed in experimentally-derived protein structures (*Jacobsen & Linderstrøm-Lang, 1949*).

The stability of a salt bridge depends on the environment around the residues, the pH, and the distance and geometric orientation between the involved residues. Salt bridges may be located on the protein surface, where they are exposed to the solvent, or they can be found buried within the hydrophobic interior of the folded protein. Studies on different protein families have found that buried salt-bridges are more likely to be conserved and functionally relevant than surface exposed ones (*Schueler & Margalit, 1995*; *Takano et al., 2000*). Surface exposed salt bridges are generally weaker, more variable in their contribution to stability, and more difficult to predict (*Sarakatsannis & Duan, 2005*). Double mutant cycles in barnase illustrate the distinction between forming a salt bridge in the low dielectric environment of the protein core versus the higher dielectric on the protein surface. These experiments show that an internal salt bridge stabilizes the protein by more than 3 kcal mol$^{-1}$ (*Vaughan et al., 2002*), whereas a surface exposed salt bridge is about 10 times weaker, ∼0.3 kcal mol$^{-1}$ at low ionic strength, a value that decreases to zero at higher ionic strength (*Serrano et al., 1990*).

Although solvent-exposed salt bridges are often observed in crystal structures of proteins, it is thus not always clear whether these interactions are stable in solution, in particular because their dynamic and transient nature make them difficult to study. With its site-specific resolution and ability to detect even transient interactions, NMR spectroscopy can, however, be used to study salt bridges in solution, and to determine the extent to which different residues interact. In one intriguing study, a range of different NMR experiments were used to examine three different potential salt bridges in the B1 domain of Protein G (hereafter called GB1) (*Tomlinson et al., 2009*). Six different crystal structures of GB1 show that three of six solvent-exposed lysines, K12, K39 and K58, form salt-bridges to the nearby acidic residues E12, E23 and D55 (*Tomlinson et al., 2009*) (Fig. 1A). To examine whether these ionic interactions are present in solution, the authors performed a series of experiments that include monitoring both the lysine nitrogen and proton chemical shifts, and the hydrogen-deuterium isotope effects on the ammonium group, while titrating the carboxylates to protonate them. Surprisingly, for two of the putative ion pairs (K12–E23 and K58–D55) the results showed essentially no observable changes at the lysines as the carboxylates changed protonation state, suggesting no substantial formation of a salt bridge.

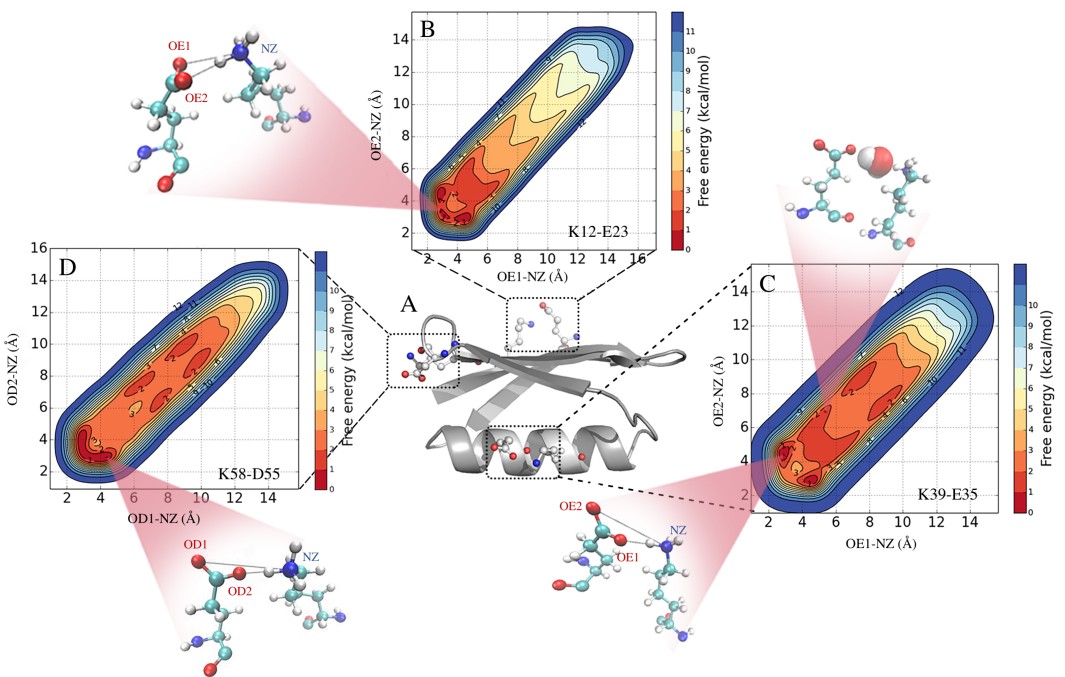

**Figure 1** **Formation and stability of salt bridges in the B1 domain of Protein G (GB1).** In (A) we show the structure of GB1 in cartoon representaion, highlighting the location of the three salt bridges that are found in crystal structures of GB1. In (B–D) we show the free energy profiles of salt bridge formation as obtained from metadynamics simulations with the CHARMM 22* force field and TIPS3P water model. The different inserts illustrate representative structures and show both contact ion pairs where the two residues are in direct interactions, and solvent-separated ion pairs where one or more water molecules sit between the charged residues.

For one pair, the intra-helical K39–E35, there was a small change in the $^{15}N$ chemical shift of the ammonium group as pH was varied to protonate E35 (and other carboxylates), though no change was observed for the $^{1}H$ chemical shift nor for the isotope effects. These observations were also supported by the determination of the pKa values of the carboxylate and ammonium groups in these residues. The authors therefore conclude that two of the salt bridges (K12–E23 and K58–D55) are not formed in solution, while the third (K39–E35) may be weakly formed (though likely at a low population).

Computational methods provide an alternative approach to study the structure and dynamics of ionic interactions in solution (*Kumar & Nussinov, 2002*). All-atom, explicit solvent molecular dynamics (MD) simulations, in particular, can be used to provide insight into the structure and energetics of biomolecules in an aqueous environment. For such simulations to provide an accurate description of salt bridges it is, however, important that the energy function (force field) used provides an accurate description of the balance between the many forces that determine salt bridge strengths. As part of the re-parameterization and validation of one of the more accurate force fields, CHARMM22*, we discovered that ionic interaction between guanidinium and acetate ions were ~10 times too strong in the original CHARMM22 force field (*Piana, Lindorff-Larsen & Shaw, 2011*).

This observation led to a re-parameterization (the "DER correction") of the partial charges in Asp, Glu and Arg side chains, to bring the interactions closer to experiment (but still slightly overestimated) (*Piana, Lindorff-Larsen & Shaw, 2011*). Building upon these ideas, Debiec and colleagues examined the interactions between analogues of both arginine, lysine and histidine side chains with carboxylates in a range of force field and water models (*Debiec, Gronenborn & Chong, 2014*). Although they found considerable variability between different force fields, the general observation from the CHARMM force fields hold true, namely that the ionic interactions between side chain analogues are overstabilized relative to experiment. Such overstabilization is not without consequence when studying protein dynamics using simulations. For example, in a study of voltage gating in potassium channels the simulations were performed with the DER correction, or the even more substantial DER2 correction, in order to be able to observe the switch in ionic interactions that are central in channel gating (*Jensen et al., 2012*). Similarly, when using free energy perturbation MD simulations to predict the change in protein stability from a mutation, the results are less accurate when the mutations involve a change in charge (*Steinbrecher et al., 2017*).

In addition to the DER correction described above, several groups have examined and modified the charges in protein force fields in order to provide a more accurate description of protein electrostatics. For example, *Jensen (2008)* updated the parameters for the charged amino acids in the OPLS force field to obtain a self-consistent set of parameters that reproduce experimental hydration free energies for side chain analogues. *Debiec et al. (2016)* used the Implicitly Polarized Charge (IPolQ) method to derive a new set of partial charges for Amber, resulting in the ff15ipq force field, and showed that these modifications substantially improved the agreement with experiments on the association between analogues of charged side chains.

Here we build on the work described above by examining the extent to which seven different fixed charge force fields capture the behaviour of the three putative salt bridges in GB1. In particular, we conducted explicit-solvent molecular simulations of GB1 using the Amber ff99SB*-ILDN, Amber ff03w, CHARMM27, CHARMM22*, OPLS_2005, AMBER ff15ipq and a99SB-*disp* force fields. In contrast to studies of side chain analogues, simulations of the salt bridges in the context of a folded protein takes into account the natural geometrical and energetic constraints imposed by the protein scaffold. To accelerate sampling and ensure convergence we used metadynamics simulations to map the free energy surface of salt bridge formation. The results reveal that also in the context of a folded protein most of these force fields overestimate the population of ionic interactions, and also provide insight into the geometry of salt bridges in proteins.

## MATERIALS AND METHODS

In order to investigate potential salt-bridging interactions between residues K12-E23, K39-E35 and K58-D55 in GB1 we performed both standard MD simulations as well as enhanced sampling simulations using well-tempered metadynamics (WT-MetaD) (*Barducci, Bussi & Parrinello, 2008*). To test the effect of the force field we used seven different force fields:

**Table 1  Simulations, force fields and salt bridge stabiliy.** The table shows the stability of each of the three salt bridges, reported as the population of structures with N–O distances shorter than 5 Å, in each of the eight different combinations of force field and water models.

| Force field | Water model | $d_{max}$ | Population | | |
|---|---|---|---|---|---|
| | | | K12–E23 | K39–D35 | K58–D55 |
| CHARMM22* | TIPS3P | 13.5 Å | 0.73 | 0.42 | 0.61 |
| CHARMM22* | TIP3P | 13.5 Å | 0.67 | 0.20 | 0.61 |
| CHARMM27 | TIPS3P | 13.5 Å | 0.83 | 0.14 | 0.75 |
| Amber ff03w | TIP4P/2005 | 13.5 Å | 0.60 | 0.48 | 0.81 |
| Amber ff99SB*-ILDN | TIP3P | 13.5 Å | 0.63 | 0.72 | 0.98 |
| OPLS AA | SPC/E | 13.5 Å | 0.65 | 0.39 | 0.98 |
| a99SB-*disp* | TIP4P-D | 13.0 Å | 0.67 | 0.25 | 0.68 |
| AMBER ff15ipq | SPC/E$_b$ | 14.0 Å | 0.096 | 0.20 | 0.02 |

Amber ff99SB*-ILDN (*Hornak et al., 2006*; *Best & Hummer, 2009*; *Lindorff-Larsen et al., 2010*), Amber ff03w (*Best & Mittal, 2010*), CHARMM27 (CHARMM22 with the CMAP correction) (*MacKerell et al., 1998*; *Mackerell, Feig & Brooks, 2004*), CHARMM22* (*Piana, Lindorff-Larsen & Shaw, 2011*), OPLS_2005 (*Banks et al., 2005*), a99SB-*disp* (*Robustelli, Piana & Shaw, 2018*) and AMBER ff15ipq (*Debiec et al., 2016*) (Table 1). Each force field was combined with its "native" water model, namely the CHARMM-specific TIP3P water model for CHARMM force fields (*MacKerell et al., 1998*) (herein called TIPS3P), the original TIP3P water model (*Jorgensen et al., 1983*) for the Amber ff99SB*-ILDN force fields, TIP4P/2005 (*Abascal & Vega, 2005*) for Amber ff03w and the SPC/E water model (*Berendsen, Grigera & Straatsma, 1987*) for OPLS. In the case of the AMBER ff15ipq force field, we used the SPC/E$_b$ water model, and the simulation box was constructed with AmberTools 17 (*Case et al., 2017*) and then converted to Gromacs format. A slightly modified version of the TIP4P-D water model (*Piana et al., 2015*) was used for the a99SB-*disp* force field (*Robustelli, Piana & Shaw, 2018*). We also examined the effect of the water model by testing CHARMM22* with standard TIP3P.

## Setup and equilibration

The simulations with the updated AMBER ff15ipq and the a99SB-*disp* force fields were performed with Gromacs version 2016.1 (*Abraham et al., 2015*), while the rest of simulations were performed with Gromacs version 4.5 (*Van der Spoel et al., 2010*), in both cases using the crystal structure of GB1 (PDB ID 1PGB; *Gallagher et al., 1994*) as starting structure. The protein was centred and solvated in a dodecahedral box with an edge length of 12 Å, and the resulting system consisted of the protein (56 residues), ~5,300 water molecules and 4 sodium ions to neutralize the total charge. The cutoff for the Van der Waals interactions was 9 Å, while the long-range electrostatic interactions were calculated using the particle mesh Ewald method with a real-space cutoff of 12 Å. For each force field and water model, we first subjected the system to 0.2 ns energy minimization, followed by a 1 ns solvent equilibration with position restraints on the protein backbone, followed by 1ns of protein equilibration. After equilibration, we performed a 10 ns simulation in the NPT ensemble (*Parrinello & Rahman, 1981*; *Parrinello & Rahman, 1982*;

*Bussi, Donadio & Parrinello, 2007*) at 298 K and 1atm pressure and calculated the average volume of the system. Finally, we selected the frame with a volume closest to this average from the second half of these 10 ns, and used this as starting point for WT-MetaD simulations in the NVT ensemble (*Bussi, Donadio & Parrinello, 2007*). The lengths of all bonds to hydrogen atoms were kept fixed using the LINCS algorithm (*Hess et al., 1997*), and simulations were performed using a 2 fs time step.

## Molecular dynamics simulations

Following equilibration, we performed a standard 100 ns unbiased MD simulation with each force field. This was followed by three 100 ns WT-MetaD simulations at 298 K using the PLUMED 1.3 plugin (*Bonomi et al., 2009*) with Gromacs 4.5 and PLUMED 2.3 plugin (*Tribello et al., 2014*) with Gromacs 2016.1, and in each simulation enhancing separately the sampling of one the three salt-bridges (K12–E23, K39–E35 and K58–D55). In metadynamics simulations, sampling is enhanced by adding a history-dependent biasing potential along one or more selected degree of freedom (collective variables, CVs). In order to sample the free energy of forming the salt bridges, we used two CVs for each of the three ion-pairs (in three different simulations). For the lysine-glutamate ion pairs we used the distance between the NZ–OE1 and NZ–OE2 atoms as CVs, and for the lysine-aspartate pair we used the distance between the NZ–OD1 and NZ–OD2 atoms. For obvious symmetry reasons, the free energy surface should be identical for the two NZ-oxygen pairs (e.g., NZ-OD1/OD2), and so the use of two CVs for each salt bridge allows us both to map the interactions with various combinations of distances to the two oxygens, but also to assess convergence via the similarity between the symmetry-related atoms. In the metadynamics simulations, the biasfactor, sigma parameter, initial height of the Gaussian hills and the deposition rate were set to 4, 0.05 Å, 0.12 kcal mol$^{-1}$ and 2 ps, respectively. In order not to enhance sampling of unrealistically long distances and to cause unfolding of GB1, we introduced a restraining potential (a soft "wall") on the CVs, using the form $E_{restr} = k(d_i - d_{max})^4$. This potential acted whenever the distance, $d_i$, was above $d_{max}$, the longest distance observed in the unbiased simulations (Table 1) and used a force constant of $k = 4.8$ kcal mol$^{-1}$ Å$^{-4}$. Finally, to test the robustness of the calculations, we also performed a 50-ns parallel-tempering metadynamics (PT-MetaD) (*Hansmann, 1997*; *Bussi et al., 2006*) simulation for the K39–E35 salt bridge using the CHARMM22* force field and TIPS3P water model. In particular, in the PT-MetaD simulations, we further enhanced the sampling with exchanges between different temperatures with a replica-exchange scheme. We used six replicas (at 294 K, 298 K, 308 K, 322 K, 337 K and 353 K) where the width of the energy distribution (of all but the "neutral" 298 K replica) was increased in a preparatory step of 10 ns as previously described (*Sutto & Gervasio, 2013*). All replicas were also subjected to an additional biasing force through metadynamics with the same parameters (initial height of the Gaussian hills, deposition rate and biasfactor) used in the WT-MetaD simulation as described above.

## Analyses

We analysed the salt bridge formation by creating two-dimensional free energy profiles for each salt bridge and for each force field, and using the two nitrogen–oxygen distances

as coordinates. To quantify the formation of each salt bridge, we calculated the fraction of time (after removing the metadynamics bias) where one of the two distances were below 5 Å.

## RESULTS

### Molecular dynamics simulations of salt bridge formation

We performed MD simulations of GB1 using eight different combinations of force fields and water models (Table 1) to examine the formation of salt bridges, and to benchmark against NMR experiments. While unbiased MD simulations showed reversible formation and breaking of the ionic interactions (Fig. 2A), we decided to use enhanced sampling metadynamics simulations (Fig. 2B) to ensure better convergence of the free energy of salt bridge formation. In both types of simulations, we find that the carboxylate and ammonium group spend time at multiple, relatively distinct sets of distances, though with differences between force fields and salt bridges (see below). Thus, for each of the eight different force field combinations we performed three metadynamics simulations, one for each salt bridge pair. In each of these 24 simulations, we simultaneously biased and monitored two CVs, corresponding to the distance between the ammonium nitrogen atom in the lysine side chain, and each of the two oxygen atoms in the carboxylate group in aspartate or glutamate. The metadynamics bias did not have any major effect on the protein stability over the course of the 100 ns simulations. For example, the average backbone RMSD was below 1.5 Å for both the unbiased and biased simulations (Fig. S1).

### Free-energy surfaces and geometry of ionic interactions

We used the metadynamics simulations to reconstruct the free energy surface for salt bridge formation along the two CVs (Fig. 1 and Fig. S2). The resulting profiles show a number of notable features. First, we find that they are highly symmetric across the "diagonal", as expected because of the equivalence of the two carboxylate oxygen atoms. Thus, while this behaviour is expected based on physical grounds, it provides a useful and independent test for convergence. Secondly, the profiles reveal a number of distinct free energy minima with depths of ~1 kcal mol$^{-1}$ and separated by small free energy barriers. For example, in the case of the K12–E23 interaction in simulations using CHARMM22* and the TIPS3P water model (Fig. 1B), three minima are visible at short distances between the two charged side chains. Two of these are symmetrically placed on either side of the diagonal and correspond to one oxygen–nitrogen distance around 3.0 Å and the other at ~4.5 Å. These minima thus correspond to a salt bridge where one oxygen is close to the nitrogen (with a proton from the ammonium group in between) and the other oxygen further away. A third minimum is also observed where both oxygen–nitrogen distances are short (~2.5 Å–3.0 Å), corresponding to a different salt bridge geometry. These three minima correspond to so called contact ion pairs where the cation and anion are in direct contact. In some force fields, in particular for the K39–E35 and K58–D55 pairs, additional minima are observed where the shortest of the two oxygen–nitrogen distances is ~6 Å–10 Å (Figs. 1C, 1D and Fig. S2). These correspond to solvent-separated ion pairs, where one or more solvent molecules

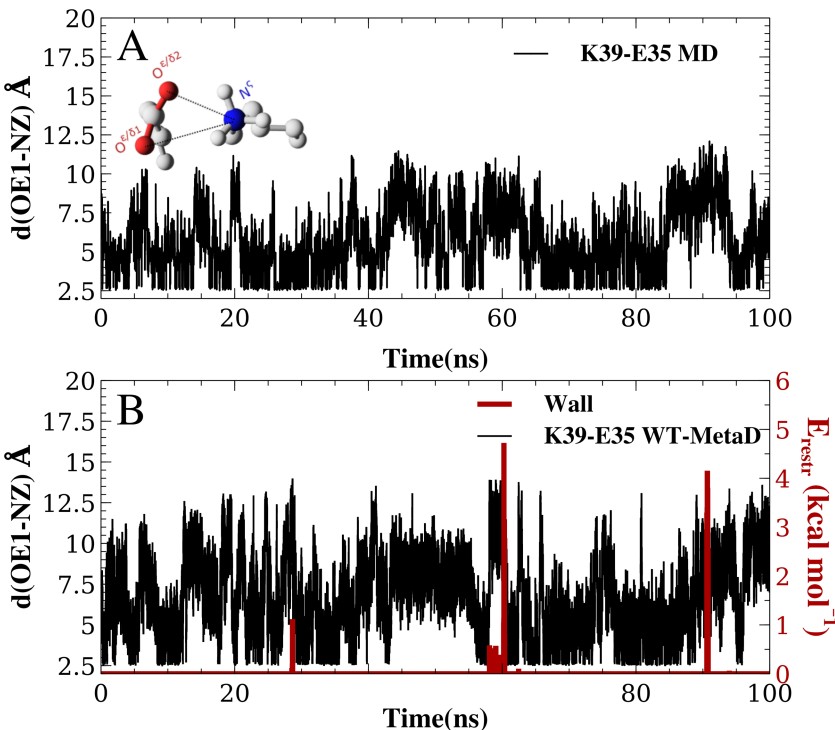

**Figure 2** **Salt bridge formation using both unbiased and enhanced sampling molecular dynamics simulations.** Using the K12–E23 salt-bridge and the CHARMM22*/TIPS3P force field as an example, we compared (A) unbiased simulations with (B) metadynamics simulations. In (B) we also show how the restraint energy acts to avoid that the ion pairs to form excessively long distances. Note that since the metadynamics simulation is biased, the resulting distribution of distances is not expected to be the same until after this bias has been removed. This unbiasing was performed before calculating the free energy profiles in all other figures and analyses.

sit between the amino acid side chain pairs and affect the electrostatic interaction of the cation and anion (*Collins, 1997*; *Marcus & Hefter, 2006*; *Zhou & Pang, 2017*).

## Comparing salt bridge stability in simulations and experiments

The free energy profiles suggest that the three salt bridges are formed and are relatively stable in all but one force field, though with variations both in geometry and the depths of the different minima. These observations appear to be in general conflict with the experimental NMR data that suggest that the K12–E23 and K58–D55 pairs do not form any substantial ionic interactions, and with at most only modest interactions between the charges in K39–E35. The major exception to these general trends is the Amber ff15ipq force field where the K12–E23 and K58–D55 salt bridges are only weakly formed, and where only the K39–E35 interaction is seen at a substantial level.

As it is difficult to calculate directly the chemical shifts and isotope effects from the simulations, we opted to compare experiments and simulations indirectly by calculating the populations of the salt bridges using the free energy surfaces. In order to be conservative and not overestimate the calculated values by including e.g., solvent mediated interactions (as it is unclear how much they contribute to the experiments), we included only conformations

where the nitrogen–oxygen distance is <5.0 Å. The resulting populations vary between force fields and salt bridges, and range from 2% to 92%, with an average value of 55% (Table 1). The salt bridge populations can also be recalculated using different definitions from the free energy profiles included as supporting data. A more detailed analysis would require a quantitative modelling of the NMR observables.

### Assessing convergence of simulations and effect of simulation parameters

When MD simulations are in apparent disagreement to experiments, it is important to assess whether these differences are due to the force field and simulation parameters, or whether they can be explained e.g., by insufficient sampling. While it is difficult to prove that simulations are converged, we performed several tests that suggest that our observations are rather robust. First, as discussed above, the high degree of symmetry of the calculated free energy surfaces suggest that the individual nitrogen–oxygen distance distributions are relatively converged. Second, we calculated the free energy of salt bridge formation during the simulation (from the populations where the nitrogen–oxygen distance is <5.0 Å) and monitored the time evolution of this (Fig. 3 shows the results from three WT-MetaD simulations with CHARMM22*/TIPS3P). While the values fluctuate for the different salt bridges, they are relatively stable in the second half of the simulations. Finally, for the K35–E39 salt bridge we also performed a parallel-tempering metadynamics simulation in CHARMM22*/TIPS3P, and compared the results to that from the WT-MetaD (Fig. 3). The two simulations converge to essentially the same free energy difference. Based on these observations we estimate an error of about 0.5 kcal mol$^{-1}$, corresponding roughly to an error of 0.2 for the population of the salt bridges when the free energy of salt bridge formation is close to zero.

Another key parameter in a simulation is the cutoff used to truncate the Lennard-Jones interactions and to switch between a direct calculation of electrostatic forces and the calculations of longer-range electrostatics using Ewald summation. For the K12–E23 and K35–E39 salt bridges and the CHARMM22*/TIPS3P force field we therefore repeated the WT-MetaD simulations varying this cutoff between 9.0 Å–14.0 Å and calculated the free energy of salt bridge formation (Fig. 4). While the results vary slightly and in accordance with the estimated uncertainty of our simulations, we find no systematic dependency of the free energy differences on the cutoff used. Thus, these calculations suggest that the cutoff used in the simulations are sufficient to obtain reasonable results.

## DISCUSSION

We have used enhance sampling simulations to analyse the stability of salt bridges in commonly used fixed charge force fields for molecular dynamics simulations. We estimate the error of the calculated free energies of salt bridge formation to be ~0.5 kcal mol$^{-1}$, corresponding to errors of the populations of ~20% (down to 10% for the most skewed populations). Surprisingly, we find that the K12–E23 and K58–D55 pairs form the most stable salt bridges in most of the force fields (~64% on average across force fields) with the K39–E35 pair being distinctly less stable (on average ~35%). These observations are in clear

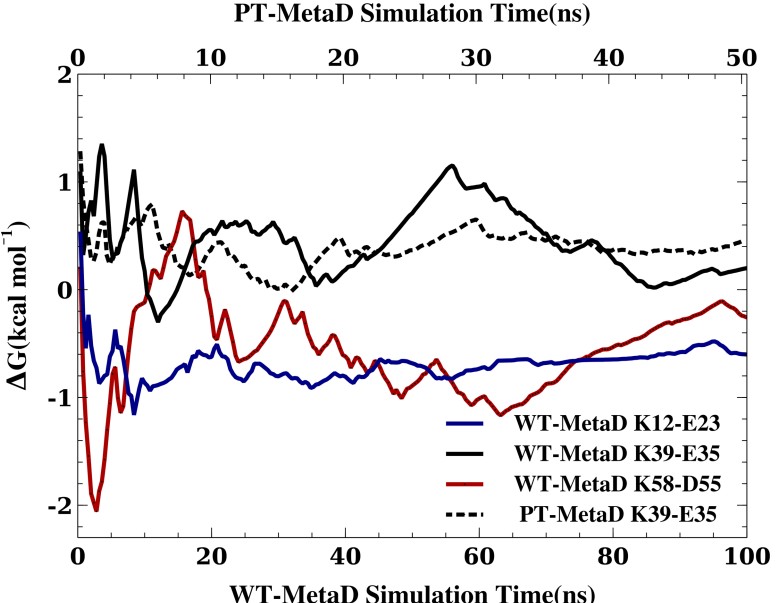

**Figure 3  Assessing the convergence of the free energy differences.** In the figure we show the free energy of salt bridge formation as a function of time, using the CHARMM22*/TIPS3P force field as an example. Initially, the values for the three different salt bridges fluctuate, but eventually converge after 100 ns of simulation (solid lines, bottom axis). We also performed a PT-MetaD simulation for the of K39–E35 salt bridge as an alternative approach to determine the free energy landscape (dashed line). After initial fluctuations, the free energy difference converges after 50 ns (top axis) to a value close to that obtained using WT-MetaD.

deviation from those expected from the experimental observations, in that the simulations generally appear both to overestimate the stability of the salt bridges, and that the order of salt bridge stability also appears to be wrongly predicted. In addition to differences in the partial charges we note that differences in bonded terms, e.g., in the torsional terms that affect side chain rotamers, can affect the relative stability and geometry of the salt bridges.

The Amber ff15ipq force field is the major outlier to the above observed trends. Thus, in contrast to the other force fields that overestimate the stability of the salt bridges, this force field is in substantially better agreement with experiment in that it finds that the K12–E23 and K58–D55 salt bridges are very weakly formed (10% and 2%, respectively). The K39–E35 salt bridge is formed ~20% of the time in the ff15ipq force field, and a more detailed comparison to the raw NMR data would be needed to determine whether this is in agreement with experiments.

For the intra-helical K39–E35 interaction we note that K39 also forms transient interactions with other residues (including E48 and N43), suggesting that the variations observed between force fields might be the cumulative effect of a number of differences between the force fields. We find that the ionic interactions are slightly stronger in TIP3PS than in standard TIP3P, in line with previous observations from model compounds (*Debiec, Gronenborn & Chong, 2014*). As also observed before (*Piana, Lindorff-Larsen & Shaw, 2011*;

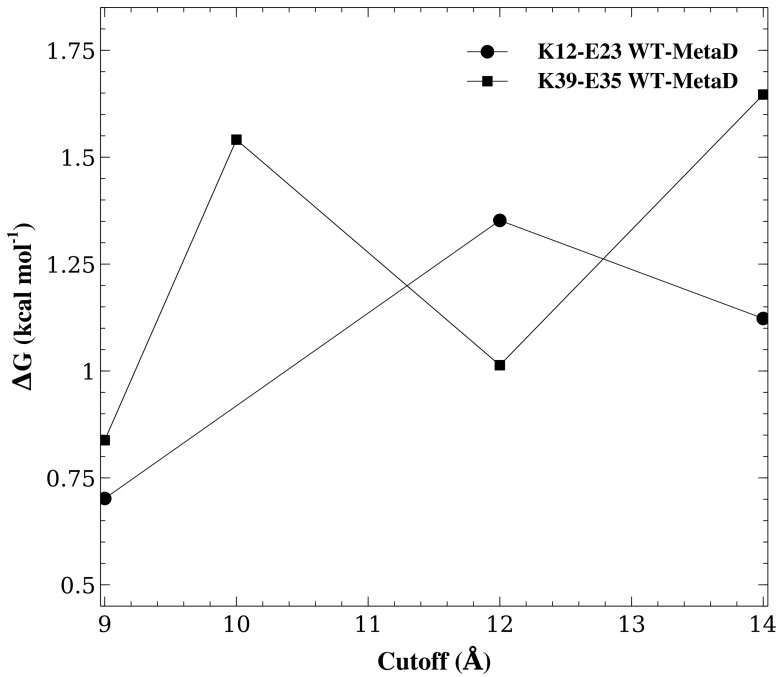

**Figure 4** **Assessing the effect of varying the cutoff distance.** We repeated the WT-MetaD for two of the salt bridges (K12–E23 and K39–E35) but varying the cutoff for the Lennard-Jones interactions and for switching from the direct calculation of electrostatic interactions to Ewald summation. The figure shows the effect of varying this cutoff on the stability of these two salt bridges.

*Debiec, Gronenborn & Chong, 2014*), we find that the DER correction in CHARMM22* decreases the strength of salt bridges to be more in line with the experiment.

## CONCLUSIONS

Ionic salt bridge interactions are pervasive in experimental protein structures and have in certain cases been shown to contribute both to protein stability (*Vaughan et al., 2002*) and fast association kinetics (*Schreiber, Haran & Zhou, 2009*). Nevertheless, our understanding of the geometry and energetics of salt bridges in solution is limited by the difficulty in experimental and computational studies. While molecular dynamics simulations in explicit solvent may in principle be a quantitative and predictive model for salt bridge formation, the accuracy of such simulations hinges upon the force fields used. Building upon earlier work on model compounds (*Piana, Lindorff-Larsen & Shaw, 2011*; *Debiec, Gronenborn & Chong, 2014*; *Debiec et al., 2016*), we have here performed simulations of GB1 to determine the free energy landscape of lysine-carboxylate salt bridge formation. In these analyses we opted to examine only fixed charge force fields, but note that at least in the case of guanidinium-acetate interactions polarizable force fields have been found to give good agreement with experimental data on salt bridge strengths in model compounds (*Debiec et al., 2016*). Our comparison with experimental NMR data on GB1 suggest that while the force fields recapitulate the transient and weak nature of

these solvent exposed ionic interactions, all but one appear to overestimate their stability slightly. This observation is in line with those from the model compounds, suggesting that together these kinds of calculations might also be used to improve the force fields. Indeed, the newly developed charge model for the Amber ff15ipq force field provides a more balanced description of salt bridge interactions for model compounds (*Debiec et al., 2016*), and our results on GB1 support this observation.

In general, we urge practitioners of MD simulations to take the small, but significant force field bias in many force fields into account when interpreting the importance of salt bridges observed in simulations, unless such observations are supported by experimental data. Finally, we hope that experimentalists will continue to develop approaches to study electrostatic interactions in proteins, in particular experiments that can be compared directly to simulations. Recent examples include extension of the GB1 studies to salt bridges in barnase (*Williamson et al., 2013*), novel NMR methods for studying electrostatics (*Hass & Mulder, 2015*), NMR methods to study arginine side chains (*Mackenzie & Hansen, 2017*; *Yoshimura et al., 2017*), and the engineering of a protein without titratable side chains as a platform for studies of protein electrostatics (*Højgaard et al., 2016*).

## ACKNOWLEDGEMENTS

We thank the members of the Lindorff-Larsen group for advice and discussions.

### Funding

Elena Papaleo and Kresten Lindorff-Larsen were supported by a Hallas-Møller stipend from the Novo Nordisk Foundation (to Kresten Lindorff-Larsen). Mustapha Carab Ahmed and Kresten Lindorff-Larsen are currently supported by the BRAINSTRUC initiative from the Lundbeck Foundation. The funders had no role in study design, data collection and analysis, decision to publish, or preparation of the manuscript.

### Grant Disclosures

The following grant information was disclosed by the authors:
Novo Nordisk Foundation.
Lundbeck Foundation.

### Competing Interests

Elena Papaleo is an Academic Editor for PeerJ.

### Author Contributions

- Mustapha Carab Ahmed performed the experiments, analyzed the data, contributed reagents/materials/analysis tools, prepared figures and/or tables, authored or reviewed drafts of the paper, approved the final draft.
- Elena Papaleo conceived and designed the experiments, performed the experiments, analyzed the data, contributed reagents/materials/analysis tools, prepared figures and/or tables, authored or reviewed drafts of the paper, approved the final draft.

- Kresten Lindorff-Larsen conceived and designed the experiments, analyzed the data, authored or reviewed drafts of the paper, approved the final draft.

## Data Availability

The raw data are provided in a Supplemental File.

## Supplemental Information

Supplemental information for this article can be found online at http://dx.doi.org/10.7717/peerj.4967#supplemental-information.

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
