# Peer review of "How well do force fields capture the strength of salt bridges in proteins?"

_PeerJ, doi:10.7717/peerj.4967_

## Round 0.1 · original submission · Major Revisions

The two reviews obtained for this manuscript are between positive and very positive, but with important suggestions and relevant criticisms that need to be addressed.

Reviewer 1 ·

Basic reporting

This paper is written in good English and easy to follow. The new work is clearly placed within the field, even including some historic contributions. Art work is relevant and well prepared.

Experimental design

The work performed is original and appears sufficiently detailed to allow reproduction, but lacks key simulations to support the claim made in the title.

Validity of the findings

Specifically, the paper deals with the interrogation of salt bridges on the surface of the protein GB1 by MD simulations with several force fields (FFs) and solvent models. Despite observations of several salt bridges in PGB1 by X-ray crystallography, NMR failed to detect the presence of stable ionic bridges, meaning that these interactions in solution are weak or absent (main conclusion from the work by Tomlinson 2009). Free energy calculations in the current work argue that MD overestimates ionic contributions with the FFs tested. This can be considered an extension of the study by Chong and co-workers (Debiec 2014) on amino acid analogues, with ditto conclusion. The authors did not address the fundamental question whether simulations in a crystal form of the protein could have explained the difference of the solution NMR and X-ray observations. This would be one way to test whether the MD force fields are any good.

A second shortcoming is that the authors have not included the newly developed ff15ipq FF, which is supposed to have mitigated problems of overemphasizing energetics of salt bridges (Debiec 2016).

Making these two important controls would allow the authors to make true their title: How well do force fields capture the strength of salt bridges in proteins? With the current data the question is still open.

Reviewer 2 ·

Basic reporting

• The overall clarity of the manuscript is high, the figures are high quality, well labeled and described, and the authors’ raw data is provided.
• The authors provide context including the importance of salt bridges and prior simulation work. However, it should be made clearer that the work described was focused on fixed-charge force fields, and the benefits and limitations of this approach briefly explained.

Experimental design

• The research described is relevant and overall well-designed, and the methods described in sufficient detail.
• In lines 237-241, the authors state that:
In order to be conservative and not overestimate the calculated values by including e.g. solvent mediated interactions (as it is unclear how much they contribute to the experiments), we included only conformations where the nitrogen–oxygen distance is < 5.0 Å.
The selected cutoff of a minimum nitrogen-oxygen distance 5.0 Å is actually generous and includes conformations in which the nitrogen and oxygen are not in direct contact. A more conservative cutoff would be 4 Å ; this will yield a decrease in the populations listed in Table 1, but should not impact the conclusions. The use of the minimum nitrogen-oxygen distance as the coordinate should also be clarified.

Validity of the findings

• The authors’ findings are robust and validated by testing of alternative simulation methods, and their conclusions are well-stated. As in the introduction, the conclusions should make clear that their findings apply to fixed-charge force fields in particular. It is worth mentioning that work on model systems, including the cited Debiec et al. 2016 paper, suggests that polarizable force fields yield salt bridge interactions in better agreement with experiment than the tested fixed-charge force fields.

Additional comments

• The authors should provide some discussion of the potential impact on their results of differences between the bonded parameters of the tested force fields, which may impact the conformations accessible to the interacting side chains of interest.
• In lines 224-228 the authors state that:
In the case of the K39–E35 and K58–D55 pairs, additional minima are observed
where the shortest of the two oxygen–nitrogen distances is ~ 6 Å – 10 Å (Figs. 1C and 1D). These correspond to solvent-separated ion pairs, where one or more solvent molecules sit between the amino acid side chains between the pairs and effecting the electrostatic interacting of the cation and anion (Collins, 1997; Marcus & Hefter, 2006; Zhou & Pang, 2017).
Based on the results shown in Supplemental Figure 2, the solvent-separated ion pair is also observed for the K12-E23 salt bridge, most clearly for CHARMM27 and AMBER ff03w. In addition, the location of the minima, in terms of the shortest oxygen-nitrogen distance, is closer to 4-7 Å rather than 6-10 Å.
• Supplemental Figure 2 refers to “Amber ff03” rather than “Amber ff03w”

---

## Round 0.2 · accepted · Accept

Dear Kresten, thank you for revising the manuscript thoroughly according to the reviewers comments. I particularly appreciate that you performed additional simulations to address the comments.

#